# Is Tirzepatide the New Game Changer in Type 2 Diabetes?

Giuseppe Lisco [1], Olga Eugenia Disoteo [2], Vincenzo De Geronimo [3], Anna De Tullio [1], Vito Angelo Giagulli [1], Edoardo Guastamacchia [1], Giovanni De Pergola [4], Emilio Jirillo [1] and Vincenzo Triggiani [1,*]

1 Interdisciplinary Department of Medicine, Section of Internal Medicine, Geriatrics, Endocrinology and Rare Diseases, University of Bari "Aldo Moro", 70124 Bari, Italy; giuseppe.lisco@uniba.it (G.L.); annadetullio16@gmail.com (A.D.T.); vitogiagulli58@gmail.com (V.A.G.); edoardo.guastamacchia@uniba.it (E.G.); emilio.jirillo@uniba.it (E.J.)
2 Unit of Endocrinology, Diabetology, Dietetics and Clinical Nutrition, Sant Anna Hospital, 22020 San Fermo della Battaglia, Italy; olgaeugenia.disoteo.amediabete@gmail.com
3 Unit of Endocrinology, Clinical Diagnostic Center Morgagni, 95100 Catania, Italy; vdg@iol.it
4 Center of Nutrition for the Research and the Care of Obesity and Metabolic Diseases, National Institute of Gastroenterology IRCCS "Saverio de Bellis", 70013 Castellana Grotte, Italy; giovanni.depergola@irccsdebellis.it
* Correspondence: vincenzo.triggiani@uniba.it

**Abstract: Background:** Tirzepatide (TZP) is a once-weekly glucagon-like peptide 1 (GLP-1) and glucose-dependent-insulinotropic-polypeptide (GIP) receptor co-agonist approved for T2D. TZP provides promising evidence in improving glucose control and weight loss in T2D and obesity across preclinical and human studies, including data from the SURPASS program. **Aims:** The goal of this paper was to review the evidence on TZP in terms of glucose control, body weight, and the progression of chronic diabetes-related complications and comorbidities. **Results:** The mean change in HbA1c ranged from −1.6% to −2.06% over placebo, from −0.29% to −0.92% over each GLP-1RAs, and from −0.7% to −1.09% over basal insulins. In SURPASS-6, TZP was more effective than insulin lispro U100 added to basal insulin in reducing $HbA_{1c}$ levels at the study end (−2.1% vs. −1.1%, respectively). Compared to placebo, TZP induces a significant weight loss: 7.5 (5 mg/week); 11 (10 mg/week); and 12 kg (15 mg/week). Compared to GLP-1RAs, TZP reduces body weight from −1.68 kg to −7.16 kg depending on the dose (5 to 15 mg, respectively). Compared to basal insulin alone rigorously titrated, TZP added onto basal-insulin results in the best strategy for the composite endpoint of improvement of glucose control and weight loss. In SURPASS-6, TZP compared to insulin lispro U100 in add-on to insulin glargine U100 reduced body weight by 9 kg in mean (versus weight gain in basal-bolus users: +3.2 kg). TZP has pleiotropic effects potentially dampening the individual cardiovascular risk, including a reduction in systolic arterial pressure by 4 to 6 mmHg and total cholesterol by 4–6% compared to baseline. A post hoc analysis of SURPASS-4 revealed that TZP, compared to glargine U100, delayed the rate of glomerular filtration decline (−1.4 mL/min vs. −3.6 mL/min, respectively), reduced the rate of urinary albumin excretion (−6.8% vs. +36.9%, respectively), and was associated with a lower occurrence of the composite renal endpoint (HR 0.58 [0.43; 0.80]). **Conclusions:** Consistent evidence indicates that TZP dramatically changes the clinical course of T2D in different clinical scenarios. The efficacy and safety of TZP on chronic diabetes-related comorbidities and complications seem promising, but ongoing trials will clarify the real benefits.

**Keywords:** tirzepatide; type 2 diabetes; glycemic control; weight loss; cardiovascular outcomes; renal outcomes; liver steatosis

## 1. Background

Over the last two decades, we observed substantial progress in the management of type 2 diabetes (T2D) and its related comorbidities from a glucocentric to a cardio-metabolic-nephron-centric treatment [1]. Current guidelines recommend a comprehensive multifactorial approach by using therapeutic strategies and evidence-based approaches to

reduce risks of microvascular, kidney, neurologic, and cardiovascular complications [2]. The sacred temple of the therapeutic paradigm of T2D now has four ionic columns, including tailored control of glucose levels, arterial pressure, lipid management, and the incorporation of medical treatment with proven cardiovascular and renal benefits [2]. Body weight control is also essential in managing T2D, and a tailored treatment is necessary to provide considerable weight loss or prevent weight gain [3].

Tirzepatide (TZP) is a once-weekly glucagon-like peptide 1 (GLP-1) and glucose-dependent-insulinotropic-polypeptide (GIP) receptor dual agonist approved for T2D in several countries. TZP binds to both GIP and GLP-1 receptors with different degrees of affinity but efficiently replaces the incretin signaling dampened in T2D [4]. TZP has been included in a novel class of antihyperglycemic agents, also known as "twincretins", with excellent results in terms of improvement of glucose control and weight loss, as indicated by preclinical studies and clinical trials [5–7], attributable to the synergistic effects of the concomitant dual agonism on GIP and GLP-1 receptors.

This review aims to encompass recent evidence on TZP in T2D, specifically highlighting the foremost effects of TZP on glucose control, body weight, and diabetes-related outcomes. Moreover, the last part of this paper focuses on safety data on TZP in T2D and obesity from clinical trials, systematic reviews, and meta-analyses. Last, ongoing trials assessing the efficacy and safety of TZP on the most relevant diabetes-related outcomes, such as chronic renal disease, atherosclerotic cardiovascular disease, heart failure, liver steatosis, and metabolic-associated fatty liver disease, are also listed.

## 2. Tirzepatide and Glucose Control

The SURPASS program of randomized clinical trials (RCTs) provided evidence of TZP efficacy and safety in several clinical settings [8], from naïve patients to uncontrolled T2D with basal-insulin regimen, as summarized in Table 1. A comprehensive overview of the risk of bias assessment of the SURPASS program trials is shown in Supplementary Materials (Figures S1 and S2, and Table S1).

In SURPASS-1 [9], TZP amounts of 5, 10, and 15 mg/week significantly reduced the levels of glycated hemoglobin (HbA1c) compared to placebo in a cohort of naïve patients recently diagnosed with T2D. More precisely, up to 92% of them achieved HbA1c levels less than 7%, considered the optimal glucose target for most, and up to 52% of participants normalized their glucose levels (HbA1c < 5.7%).

In SURPASS-2 [10], TZP has demonstrated superiority over 1 mg of semaglutide in terms of HbA1c reduction, with 92% of patients obtaining the optimal glucose target (i.e., HbA1c < 7%), and 51% normalized their glucose control, as indicated by HbA1c levels less than 5.7%.

In SURPASS-3 [11], TZP was demonstrated to be superior to insulin degludec U100, as up to 93% of individuals randomized to TZP had HbA1c < 7% and up to 48% had HbA1c levels less than 5.7% at the trial end.

In SURPASS-5 [12], TZP significantly improved glucose control in add-on to insulin glargine U100 over glargine U100 alone titrated with a rigorous treat-to-target approach. Up to 97% of patients had HbA1c < 7% and up to 62% normalized their glucose levels (HbA1c < 5.7%).

The cardiovascular safety trial SURPASS-4 [13] showed that TPZ, compared to a rigorously titrated regimen of insulin glargine U100, significantly improved glucose control in T2D patients with high CV risk. Also, TZP provided considerable weight loss and lower frequency and severity of hypoglycemic episodes with CV effects comparable to that of insulin glargine U100.

Most importantly, TZP was demonstrated to be superior to comparators at all doses (5, 10, and 15 mg/week). The mean change in HbA1c ranged from −1.6% to −2.06% over placebo, from −0.29% to −0.92% over each GLP-1 receptor agonist (GLP-1RA), and from −0.7% to −1.09% over basal insulins [14].

**Table 1.** Summary of the SURPASS program, Phase III randomized clinical trials in T2D.

| Study (Year) | Study Duration (Weeks) | Intervention vs. Comparator | Randomization (Number of Participants) | Baseline Characteristics | Main Findings |
|---|---|---|---|---|---|
| SURPASS-1 (2021) | 40 | TZP vs. placebo (monotherapy in naïve T2D patients) | 1:1:1:1 TZP 5 mg qw (125) TZP 10 mg qw (125 pz) 15 mg qw (125 pz) Placebo qw (121 pz) | HbA$_{1c}$ 7.9% Mean age 54 yrs Women 48% Diabetes duration 4.7 yrs BMI 31.9 kg/m$^2$ | Mean change from baseline in HbA$_{1c}$: −1.87% with TZP 5 mg −1.89% with TZP 10 mg −2.07% with TZP 15 mg +0.04% with placebo |
| | | | | | % of participants achieving HbA$_{1c}$ < 7%: 87–92% with TZP 5–15 mg 20% with placebo |
| | | | | | % of participants achieving HbA$_{1c}$ ≤ 6.5%: 81–86% with TZP 5–15 mg 10% with placebo |
| | | | | | % of participants achieving HbA$_{1c}$ < 5.7%: 31–52% with TZP 5–15 mg 1% with placebo |
| | | | | | Mean change in weight from baseline: −7 to −9.5 kg with TZP 5–15 mg |
| SURPASS-2 (2021) | 40 | TZP vs. semaglutide (T2D patients with poor glycemic control while on metformin) | 1:1:1:1 TZP 5 mg qw (470 pz) TZP 10 mg qw (469 pz) TZP 15 mg qw (470 pz) Semaglutide 1 mg qw (469 pz) | HbA$_{1c}$ 8.28% Mean age 56.6 yrs Women 53% Diabetes duration 8.6 yrs BMI 31.9 kg/m$^2$ Waist 109.3 cm eGFR 96 mL/min | Mean change from baseline in HbA$_{1c}$: −2.01% with TZP 5 mg −2.24% with TZP 10 mg −2.3% with TZP 15 mg −1.86% with semaglutide 1 mg |
| | | | | | % of participants achieving HbA$_{1c}$ < 7%: 82–86% with TZP 5–15 mg 79% with semaglutide 1 mg |
| | | | | | % of participants achieving HbA$_{1c}$ ≤ 6.5%: 69–80% with TZP 5–15 mg 64% with semaglutide 1 mg |
| | | | | | % of participants achieving HbA$_{1c}$ < 5.7%: 27–46% with TZP 5–15 mg 19% with semaglutide 1 mg |
| | | | | | Mean change in weight from baseline: −7.6 to −11.2 kg with TZP 5–15 mg −5.7 kg with semaglutide 1 mg |

**Table 1.** *Cont.*

| Study (Year) | Study Duration (Weeks) | Intervention vs. Comparator | Randomization (Number of Participants) | Baseline Characteristics | Main Findings |
|---|---|---|---|---|---|
| SUPRASS-3 (2021) | 52 | TZP vs. insulin degludec U100 (add-on to metformin +/− sodium-glucose transporter 2 inhibitors) | 1:1:1:1 TZP 5 mg qw (361 pz) TZP 10 mg qw (361 pz) TZP 15 mg qw (361 pz) Insulin degludec U100 (361 pz) | HbA$_{1c}$ 8.17% Age $\geq$ 18 yrs Diabetes duration 8.4 yrs BMI > 25 kg/m$^2$ | Mean change from baseline in HbA$_{1c}$: −1.9% TZP 5 mg −2.2% TZP 10 mg −2.37% TZP 15 mg −1.34% insulin degludec U100 % of participants achieving HbA$_{1c}$ < 7%: 82–93% with TZP 5–15 mg 61% with insulin degludec U100 Mean change in weight from baseline: −7.5 to −11.2 kg with TZP 5–15 mg +2.3 kg with insulin degludec U100 |
| SURPASS-4 (2021) | 52 | TZP vs. insulin glargine U100 (add-on to metformin +/− secretagogues +/− sodium-glucose transporter 2 inhibitors in any combinations in patients at high cardiovascular risk) | 1:1:1:3 TZP 5 mg qw (329 pz) TZP 10 mg qw (328 pz) TZP 15 mg qw (338 pz) Insulin glargine U100 (1000 pz) | HbA$_{1c}$ 8.5% Age $\geq$ 18 yrs Diabetes duration 11.8 yrs BMI > 25 kg/m$^2$ | Mean change from baseline in HbA$_{1c}$: −2.43% with TZP 10 mg −2.28% with TZP 15 mg −1.44% with insulin glargine U100 MACE-4 events (cardiovascular death, myocardial infarction, stroke, hospitalization for unstable angina): Hazard ratio = 0.74 (95% CI 0.51–1.08) |
| SURPASS-5 (2022) | 40 + 4 (4-week safety follow-up) | TZP vs. placebo (add-on to insulin glargine U100 of at least 20 IU per day or 0.25 IU/kg +/− metformin of at least 1500 mg per day) | 1:1:1:1 TZP 5 mg qw (116 pz) TZP 10 mg qw (119 pz) TZP 15 mg qw (120 pz) Placebo (120 pz) | HbA$_{1c}$ 8.3% Age $\geq$ 18 yrs BMI > 23 kg/m$^2$ Diabetes duration 13.3 yrs Mean insulin dose 30 IU/day Women 44% | Mean change from baseline in HbA$_{1c}$: −2.11% TZP 5 mg −2.40% TZP 10 mg −2.34% TZP 15 mg −0.86% placebo % of participants achieving HbA$_{1c}$ < 7%: 88%, pooled of TZP (all doses) 34.5% with placebo % of participants achieving HbA$_{1c}$ $\leq$ 6.5%: 79%, pooled of TZP (all doses) 17.3% with placebo % of participants achieving HbA$_{1c}$ < 5.7%: 38%, pooled of TZP (all doses) 3% with placebo Mean change in weight from baseline: −5.4 to −8.8 kg with TZP 5–15 mg +1.6 kg with placebo |

**Table 1.** *Cont.*

| Study (Year) | Study Duration (Weeks) | Intervention vs. Comparator | Randomization (Number of Participants) | Baseline Characteristics | Main Findings |
|---|---|---|---|---|---|
| SURPASS-6 (2022) | 52 | TZP vs. insulin lispo U100 (add-on to basal-insulin glargine U100 +/− metformin of at least 1500 mg per day, sulphonylureas, or dipeptidyl peptidase type IV inhibitors) | 1:1:1:3 TZP 5 mg qw (243 pz) TZP 10 mg qw (238 pz) TZP 15 mg qw (236 pz) Insulin lipro U100 (708 pz) | HbA$_{1c}$ 8.8% Age $\geq$ 18 yrs BMI > 23 kg/m$^2$ eGFR > 30 mL/min Women 57% Diabetes duration 14 yrs Median insulin dose 46 IU (0.53 IU/kg) | Mean change from baseline in HbA$_{1c}$: −1.9% with TZP 5 mg −2.2% with TZP 10 mg −2.3% with TZP 15 mg −1.11% with insulin lispro U100 thrice a day |
| | | | | | % of participants achieving HbA$_{1c}$ < 7%: 68%, pooled TZP (all doses) 36% with insulin lispro U100 thrice a day |
| | | | | | % of participants achieving HbA$_{1c}$ $\leq$ 6.5%: 56%, pooled TZP (all doses) 22% with insulin lispro U100 thrice a day |
| | | | | | % of participants achieving HbA$_{1c}$ < 5.7%: 18%, pooled TZP (all doses) 3% with insulin lispro U100 thrice a day |
| | | | | | Mean change in weight from baseline: −6.7 to −11 kg with TZP 5–15 mg +3.2 kg with insulin lispro U100 thrice a day |

Abbreviations: TZP = tirzepatide; HbA$_{1c}$ = glycated hemoglobin; BMI = body mass index; IU = international units; MACE = major adverse cardiovascular events; qw = every week; yrs = years; +/− = with or without

In SURPASS-6, TZP was compared to insulin lispro U100 as an add-on to basal insulin (glargine U100) in patients with uncontrolled T2D [15]. After 52 weeks of treatment, HbA1c levels were lower in patients intensified with TZP than insulin lispro U100 (−2.1% vs. −1.1%, respectively), with TZP as an add-on to glargine U100 performing better than basal-bolus insulin regimen (HbA1c at the study end: 6.7% vs. 7.7%, respectively).

Overall, TZP provides additional evidence supporting the choice of incretin-based antihyperglycemic agents as the first injectable treatment for most with T2D over basal insulin and more composite insulin regimens, such as basal-plus and basal-bolus [16].

## 3. Tirzepatide and Body Weight

The results of two meta-analyses showed that TZP induces a significant weight loss, compared to placebo by around 7.5 (5 mg/week), 11 (10 mg/week), and 12 kg (15 mg/week) [17,18]. Compared to GLP-1RAs, TZP reduces body weight from −1.68 kg to −7.16 kg depending on the dose (5 to 15 mg, respectively) [14]. Compared to basal insulin alone rigorously titrated using a treat-to-target approach, TZP added onto basal-insulin results in the best strategy to improve the study outcome with a composite endpoint of improved glucose control and weight loss [19]. The SURPASS-6 found similar results, in which TZP in add-on to insulin glargine U100 reduced body weight by 9 kg in mean (versus weight gain in basal-bolus users: +3.2 kg) [15]. So far, no direct comparisons have been analyzed between TZP and sodium-glucose (co)transporter type 2 inhibitors (SGLT2is). SGLT2is were proven to induce mild-to-moderate weight loss. According to the results of one systematic review and network metanalysis, SGLT2is cause a body weight reduction of 1.3 to 3.1 kg, depending on patients' characteristics and background antihyperglycemic treatment, without a statistically significant difference between the molecules [20].

According to the results of indirect comparisons among all anti-hyperglycemic classes, including TZP, on body weight in T2D, TZP resulted in the most significant reduction in body weight with an estimated mean difference of −8.6 kg [21].

Impressive findings on weight loss have also been confirmed in obese individuals who were treated with TZP, irrespective of background diabetes. In the SURMOUNT-1 trial, the mean change in weight from baseline after 72 weeks of treatment was −15% (5 mg), −19.5% (10 mg), and −20.9% (15 mg). Body weight reduction was significantly more than placebo (−3.1%). Patients lost more than 20% of baseline body weight more frequently on TZP with 10 mg (50%) and 15 mg (57%) than on placebo (3%) [22]. The results of the SURMOUNT-2 trial confirmed important findings in patients with obesity and T2D [23]. TZP amounts of 10 and 15 mg induce more weight loss with a comparable safety profile also compared to high-dose single GLP-1RAs, such as daily semaglutide 0.4 mg, weekly semaglutide 2.4 mg, and daily liraglutide 3 mg, as a network meta-analysis of randomized clinical trials has recently found [24].

Although TZP is not formally approved for obesity, current evidence shows results never seen before with other pharmacological approaches. Starting from these data and given the results of ongoing trials that will be published in the near future, TPZ is expected to readdress the role of medical compared to surgical (bariatric) treatment in the management of comorbid obesity [25].

## 4. Tirzepatide and Cardiovascular Outcomes

Cardiovascular (CV) diseases (CVD), especially coronary disease and stroke, affect approximately 30% of patients with T2D and represent the most common cause of mobility and mortality in T2D [26].

While megatrials indicated that intensive treatment of hyperglycemia resulting in more stringent glucose control is associated with a considerable reduction of microvascular complication, less clear is the relation between optimal glucose control and improvement of CV outcomes in T2D [27–30]. Evidence indicates that intensive multifactorial intervention targeting glucose, arterial pressure, lipid control, and behavior modification has sustained

beneficial effects in terms of CV prevention, CV and all-cause mortality, thus resulting in the best management of CV prevention and CVD in T2D [31].

More recently, the results of CV outcome trials have indicated that specific GLP-1RAs and SGLT2is have proven CV benefits beyond the glycemic effect. Liraglutide, compared to placebo, reduced the risk of occurrence of the primary composite endpoint (death from cardiovascular causes, nonfatal myocardial infarction, or nonfatal stroke), CV mortality, and all-cause mortality by 13%, 22%, and 15%, respectively [32]. Once-weekly semaglutide, compared to placebo, reduced the risk of nonfatal myocardial infarction by 26% (statistically significant for non-inferiority) and nonfatal stroke by 39% [33]. In PIONEER-6, oral semaglutide compared to placebo reduced the CV and all-cause mortality by 51% and 49%, respectively [34].

Empagliflozin was demonstrated to reduce the primary composite outcome of CV mortality, nonfatal myocardial infarction, or nonfatal stroke by 14% compared to placebo. The result was mostly driven by a considerable reduction in the rate of deaths for CV events by 38%; moreover, the use of empagliflozin was associated with a substantial reduction in the relative risk of hospital admission for decompensated heart failure (−35%) and all-cause mortality (−32%) [35]. In the CANVAS trial, canagliflozin reduced the primary outcome by 14%, resulting in statistically significant superiority over placebo [36]. Dapagliflozin, compared to placebo, reduced the composite endpoint of CV death and hospital admission for heart failure by 17%, which reflected a significantly lower rate of hospitalization due to decompensated heart failure (−37%) [37].

The cardiovascular outcome trial SURPASS-CVOT is ongoing to address the CV safety of TZP compared to dulaglutide in T2D individuals at high CV risk [38]. As is known, dulaglutide has been demonstrated to reduce the primary composite endpoint of non-fatal myocardial infarction, non-fatal stroke, or death from cardiovascular causes by 12% in a population of T2D on primary and secondary CV prevention, as the REWIND trial demonstrated [39]. Post hoc analyses have been carried out to address the CV safety of TZP. Sattar meta-analyzed CV data from the SURPASS program, finding that TZP, compared to placebo, had a neutral effect on cardiovascular and all-cause death and composite CV outcome [40].

Nonetheless, TZP has important pleiotropic effects, anticipating CV benefits based on the impact of TZP on surrogate outcomes. For example, arterial pressure control is associated with a significant reduction of new onset and progression of microvascular complication and ischemic stroke in T2D, with most of the benefit achieved with systolic and diastolic arterial pressure < 130/80 mmHg [41]. It has been found that TZP, over background antihypertensive treatment and statins, reduces systolic arterial pressure by 4 to 6 mmHg (5 mg to 15 mg/week) and total cholesterol by 4–6% compared to base-line [42], also improving specific signs of insulin sensitivity, such as liver steatosis and waist circumference [43].

Although there are no direct comparisons between TZP and antihyperglycemic agents with proven CV benefits in terms of improvement of arterial pressure and cholesterol levels, it should be taken in mind that SGLT2is with proven CV protection in add-on to background antihypertensive treatment are associated with a mild reduction of systolic and diastolic arterial pressure by 2.5 and 1.4 mmHg, respectively [44]. Moreover, SGLT2is induce a mild improvement or neutral effect on lipid profile when added to hypocholesterolemic agents [45–47]. Nevertheless, better results have been found in patients with fatty liver disease, in which SGLT2is reduce both total cholesterol by 2.7 mg/dL and triglycerides by 16.8 mg/dL [48]. GLP-1RAs with proven CV benefits, compared to SGLT2is, induce a similar decrease in both systolic and diastolic arterial pressure, which are clinically insignificant [49], with a modest improvement or neutral effect on lipid profile [50,51].

Overall, data on surrogate endpoints indicate that TZP could have a relevant thera-peutical potential in the prevention of CV diseases in T2D, even if specific and event-driven trials are needed to confirm this hypothesis.

### 5. Tirzepatide and Renal Outcomes

Chronic kidney disease (CKD) is one of the most common chronic complications in T2D. Up to 40% of individuals with diabetes have chronic renal impairment [52], with a wide range of renal damage, as defined by the Kidney Disease Improving Global Outcomes classification [53]. CKD represents the leading cause of end-stage renal disease and death for renal causes in T2D [54,55].

As above mentioned for CVD, antihyperglycemic drugs with proven renal benefits are desirable to affect the natural history of renal impairment associated with T2D. Compared to placebo, liraglutide has been demonstrated to reduce the composite endpoint of persisting macroalbuminuria, doubling serum creatinine level, end-stage renal disease, or death for renal disease by 26%. This finding was mostly driven by a significant reduction in the relative risk of new onset macroalbuminuria (−26%) compared to placebo [56]. Dulaglutide, compared to placebo, reduced the primary composite endpoint of first occurrence of new macroalbuminuria, sustained decline in eGFR $\geq$ 30% from baseline, or chronic renal replacement therapy by 15%. As observed for liraglutide, dulaglutide reduced the relative risk of new onset of macroalbuminuria by 23% in T2D [57]. Secondary analyses from the SUSTAIN-6 trial found that semaglutide reduced the risk of new onset or deterioration of chronic renal disease, as the composite outcome of persisting macroalbuminuria, doubling of serum creatinine, or end-stage renal disease was reduced by 36% [33]. The result of a systematic review and meta-analysis indicates that GLP-1RAs, compared to placebo, provide a statistically relevant improvement in albuminuria by around 16% [58].

SGLT2is provided consistent evidence to improve significantly renal outcomes regardless of glucose control and the presence of T2D. The CREDENCE trial was the first renal outcome trial to be designed and published for estimating the role of canagliflozin in the prevention of renal outcomes in T2D. Canagliflozin, compared to placebo, reduced the relative risk of the primary outcome (composite of end-stage renal disease, a doubling of the serum creatinine level, or death from renal or cardiovascular causes) by 30% and the relative risks of mortality for renal causes or end-stage renal diseases by 34% and 32%, respectively [59]. Dapagliflozin was found to reduce the primary composite outcome of sustained decline in the GFR $\geq$ 50%, end-stage kidney disease, or death from renal or cardiovascular causes by 44%. Interestingly, the protective effect of dapagliflozin was seen in both T2D and non-diabetic individuals [60]. In the EMPA kidney trial, empagliflozin was found to improve the primary study outcome, a composite of end-stage kidney disease, a sustained decrease in eGFR to <10 mL/min/1.73 m$^2$, a sustained reduction in the eGFR of $\geq$40% from baseline, or death from renal or CV causes, by 28% compared to placebo [61]. The result was in line with another pooled analysis in which SGLT2is were found to reduce albuminuria by 26.2% [62]. SGLT2is have demonstrated to reduce the composite kidney outcome (development of new-onset macroalbuminuria, decline in glomerular filtration rate, progression to end-stage kidney disease, or death attributable to kidney causes) by 14% to 21% [63–65].

Finerenone, a nonsteroidal, selective mineralocorticoid receptor antagonist, was found to improve renal outcomes in patients with T2D and chronic renal disease on top of renin–angiotensin system blockers. In the FIDELIO trial, finerenone reduced the relative risk of the primary outcome (composite of kidney failure, a sustained decrease in the eGFR $\geq$ 40% from baseline, or death from renal causes) by 18% in T2D individuals with established diagnosis of CKD [66]. Similar results were found by Pitt et al. in the FIGARO trial. In this trial, the cardiovascular and renal benefits of finerenone were assessed in T2D patients with a broader range of renal function (eGFR 25–90 mL/min/1.73 m$^2$). Finerenone improved the renal composite outcome by 13% compared to placebo, but the result was not statistically significant [67].

No specific trials have been conducted so far to assess the renal effect of TZP. A post hoc analysis of SURPASS-4 revealed that TZP compared to glargine U100 delayed the mean rate of glomerular filtration decline (−1.4 mL/min vs. −3.6 mL/min, respectively), reduced the mean rate of urinary albumin excretion (−6.8% vs. +36.9%, respectively),

and was associated with a lower occurrence of the composite renal endpoint by 42% (HR 0.58 [0.43; 0.80]) [68]. Despite benefits being more evident in patients with baseline renal impairment, the protective effects of TZP were also observed in normoalbuminuric and those with glomerular filtration higher than 60 mL/min [69]. To confirm positive results on renal outcomes, the meta-analysis by Mima et al. found that TZP remarkably reduced the risk of composite renal outcome by 45% and worsening albuminuria by 62% [70].

Ongoing investigation is needed to clarify the potential effect of TZP on renal outcomes in T2D.

## 6. Safety

Compared to placebo and insulin treatment, gastrointestinal adverse effects are commonly observed among TZP users [71]. Nausea was observed in 20% of TZP users, with a relative risk of 2.9 compared to comparators. Vomiting was observed in 9%, with a similar relative risk as above described for nausea (2.7). Diarrhea has been reported in 16% of TZP users (relative risk of 2 compared to comparators), while stypsis was less frequently found (2.5% with a relative risk of 3.1). Loss of appetite and dyspepsia were described in 9.6% and 7% of patients, respectively, equivalent to a relative risk of 3 and 2, respectively [72]. Nevertheless, according to the result of a meta-analysis, the overall risk of hypoglycemia and discontinuation of treatment were higher in TZP users compared to placebo, GLP-1RA, and basal-insulin (degludec U100 and glargine U100). The risk of discontinuation due to adverse events was higher in high-dose TZP than GLP-1RAs, as the risk was almost doubled (risk ratio 1.75) with 10 mg of TZP and doubled (risk ratio 2.03) with 15 mg of TZP. Hypoglycemia was more frequently reported with TZP with 15 mg than GLP-1RAs (risk ratio 3.83) but less frequently with TZP than insulin glargine (pooled risk ratio 0.4) and degludec (pooled risk ratio 0.21) [71]. The frequency of injection-site reactions is similar in TZP than GLP-1RA and insulin users, and the occurrence of any adverse event is dose-dependent, but it is usually lessened by adequate titration [73].

Overall, the frequency and severity of adverse events were comparable to those previously reported with GLP-1RAs. Severe or serious adverse events, including acute pancreatitis, cholelithiasis, biliary lithiasis, and acute cholecystitis, were similar between TZP and other classes of medications such as GLP-1RAs, insulin, or placebo [74].

As another issue, TZP was found to increase heart rate compared to other agents, including GLP-1RAs. The weekly 15 mg dose had the most significant effect on increasing heart rate compared with lower doses of TZP and other treatments. The clinical relevance of this phenomenon is still unknown [75].

## 7. Ongoing Trials

As mentioned above, chronic-diabetes-related complications represent the most common causes of comorbidities, overall and specific mortality (e.g., for CV and renal causes), impairment of the quality of life, and heartcare costs. Guidelines indicate that comprehensive management of multiple risk factors and the incorporation of drugs with proven CV and renal benefits are cost-effective in T2D.

A special mention is necessary for the metabolic-associated steatotic liver disease (MASLD), which is expected to be the most common liver disorder in the XXI century. MASLD, previously named metabolic-associated fatty liver disease (MAFLD) [76], is expected to affect billion of people worldwide with a global prevalence of around 30% [77]. MASLD is more frequently diagnosed in patients with T2D, obesity, and other common metabolic disorders, and represents a considerable risk factor for CV, renal, and liver diseases, as well as malignancies, including liver, colon, and breast cancers [78,79]. To complicate the scenario, it should be considered that MASLD is significantly underdiagnosed, especially because of diagnostic challenges. As a second issue, MASLD is a progressive disease leading to chronic fibrosis, cirrhosis, and liver cancer, and no current pharmacological approaches have been proven to delay or revert its progression [80]. So far, behavior interventions aiming to significantly reduce body weight, prevent weight gain, and maintain

weight loss over time are desirable in the treatment of MASLD and prevention of related complications at the liver and systemic sites.

TZP has, therefore, the potential to significantly improve the natural history of diabetes, and related complications and comorbidities including MASLD, thus becoming another important candidate for treatment intensification in T2D. More investigations are ongoing to address the role of TZP in some specific conditions, such as CVD, heart failure, renal impairment, MASLD, and steatohepatitis (Table 2).

**Table 2.** Summary of ongoing investigations focusing on TZP and diabetes-related chronic complications.

| Ongoing Trial (ClinicalTrial.Gov ID) | Sponsor | Kind of Study | Location(s) | Focus | Intervention/ Comparator | Start Date | Completion Date |
|---|---|---|---|---|---|---|---|
| NCT05536804 | Eli Lilly and Company | Phase II 56-week RCT | USA, Austria, Canada, Denmark, Mexico, Netherlands | Chronic kidney disease in T2D | TZP vs. placebo | 08 Feb 2023 | 26 Feb 2026 |
| NCT04166773 | Eli Lilly and Company | Phase II RCT | USA, France, Italy, Japan, Mexico, Poland, Spain, UK | Nonalcoholic steatohepatitis | TZP vs. placebo | 19 Nov 2019 | 07 Feb 2024 |
| NCT05751720 | Independent | Phase II nonrandomized, uncontrolled intervention trial | UAE (Abu Dhabi University) | Nonalcoholic fatty liver disease | TZP | Apr 2023 | Feb 2025 |
| NCT04255433 | Eli Lilly and Company | SURPASS-CVOT, Phase III, active comparator RCT | USA, Argentina, Australia, Austria, Belgium, Brazil, Canada, China, Czech Republic, France, Germany, Greece, Hungary, India, Israel, Italy, Japan, Republic of Korea, Mexico, Netherlands, Poland, Puerto Rico, Russian Federation, Slovakia, Spain, Sweden, Taiwan, Turkey, Ukraine, UK | Major cardiovascular events in high-risk T2D | TZP vs. dulaglutide | 29 May 2020 | 17 Oct 2024 |
| NCT05708859 | Independent | Phase IV RCT | USA (Torrance, California) | Progression of coronary atherosclerosis by using multidetector computed tomography | TZP vs. placebo | Dec 2023 | May 2026 |
| NCT04847557 | Eli Lilly and Company | Phase III 52-week RCT | USA, Brazil, China, India, Israel, Mexico, Puerto Rico, Russian Federation, Taiwan | Heart failure with preserved ejection fraction in obesity | TZP vs. placebo | 20 Apr 2021 | 30 Jul 2024 |

Abbreviations: RCT = randomized clinical trial; TZP = tirzepatide; USA = United States of America; UAE = United Arab Emirates; UK = United Kingdom.

## 8. Conclusions

GLP-1RAs plus basal insulins have demonstrated non-inferiority compared to more composite insulin regimes, including basal-plus and basal-bolus, in terms of glucose control. Moreover, they significantly reduce the risk of hypoglycemia and unnecessary weight gain and improve patients' adherence to prescriptions due to an easy handling of therapy. Hence, the basal-GLP-1RA regimen is considered an effective and safe therapeutic alternative to

composite insulin treatment, which is no longer considered the most effective treatment in T2D.

The SURPASS-6 results add further pieces of evidence related to a novel class of antihyperglycemic drugs, namely twincretins, for which TZP is the ancestor. Dual agonists are expected to produce remarkable innovation within the therapeutic paradigm of T2D, since combining TZP with basal insulin is better than the use of more composite insulin regimens in terms of tailored management of T2D, as recommended by guidelines, with acceptable data on safety and potential positive results in terms of prevention of chronic complications and additional conditions with a high impact on T2D patients' health.

TZP improves patients' quality of life [81,82], thus achieving another important target of the medical management of T2D. Improvement in quality of life is more significant in younger patients (i.e., <65 years) and those with baseline BMI > 25 kg/m$^2$ [83] with the achievement of multiple targets in T2D, such as lower HbA1c targets, higher body weight loss, treatment with simple-to-use and well-tolerated medications and devices, being probably the drivers of this amelioration [84].

Further evidence will be provided, better indicating the role of TZP in the prevention and treatment of CV, renal, and other metabolic disorders related to T2D.

**Supplementary Materials:** The following supporting information can be downloaded at: https://www.mdpi.com/article/10.3390/endocrines5010005/s1, Figure S1: Risk of bias assessment of randomized clinical trials of the SURPASS program. The assessment was targeted to the primary outcome (mean HbA1c change from baseline for all trials) with an intention-to-treat effect. Figure S2: Graphical representation of the risk of bias assessment of randomized clinical trials of the SURPASS program. The assessment was targeted to the primary outcome (mean HbA1c change from baseline for all trials) with an intention-to-treat effect. Table S1: Data sheet of the risk of bias assessment of randomized clinical trials of the SURPASS program.

**Author Contributions:** G.L. conceived the review; G.L. drafted the manuscript; G.L., O.E.D. and V.D.G. searched databases for original papers, reviews, systematic reviews, and meta-analyses; G.L., O.E.D., V.D.G., A.D.T. and E.G. selected appropriate references; O.E.D., V.D.G., V.A.G., G.D.P., E.G., E.J. and V.T. provided feedback. All authors have read and agreed to the published version of the manuscript.

**Funding:** This research received no external funding.

**Institutional Review Board Statement:** Not applicable.

**Informed Consent Statement:** Not applicable.

**Data Availability Statement:** Not applicable.

**Conflicts of Interest:** The authors deny any relevant competing interests related to the manuscript preparation.

## List of Abbreviations

| | |
|---|---|
| Cardiovascular | CV |
| Cardiovascular disease | CVD |
| Chronic kidney disease | CKD |
| GLP-1 receptor agonists | GLP-1RAs |
| Glucagon-like peptide 1 | GLP-1 |
| Glucose-dependent insulinotropic polypeptide | GIP |
| Glycated hemoglobin | HbA1c |
| Randomized clinical trials | RCTs |
| Sodium-glucose (co)transporter type 2 inhibitors | SGLT2is |
| Tirzepatide | TZP |
| Type 2 diabetes | T2D |

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
