# Peer review of "Is Tirzepatide the New Game Changer in Type 2 Diabetes?"

_endocrines, doi:10.3390/endocrines5010005_

Round 1

Reviewer 1 Report

Comments and Suggestions for Authors

Interesting, well written and well summerazied rewiew on the double antagonist GLP GIP (tirzepatide).

Most readers (others than those involed in clinical trials and clinical research) should be intrested since the molecule is not yet available in most european countries.

No major concerns in this format of review, but presentation  of table 1 (summary of the SURPASS program) have to be improved ! not readable nor understandable !

 GRADE method  on available results would be a plus !

Comments on the Quality of English Language

No major concern.

Author Response

Dear Reviewer, 
Thanks for your comments. 

I revised Table 1 to improve its readability, as you indicated. Please check the text for details (changes are in red).

GRADING the evidence requires significantly more time, as specific questions should be assessed point-by-point for each outcome and end-point (i.e., grading the effect on HbA1c, body weight, achievement of glycemic target, etc). Moreover, please consider that it is a review, not a guideline. What you requested could be a starting point for a specific systematic review/meta-analysis or, even more, a position statement when real-life data will also be available. Nevertheless, we assessed the risk of bias of the SURPASS RCTs to better cope with your request for this paper. You can find graphics and the data sheet in the supplemental material.

Reviewer 2 Report

Comments and Suggestions for Authors

1. Tirzeptide is a GLP-1/GIP dual agonist, but not a GLP-1 receptor agonist. Please avoid the confusion regarding the two drug classes. 

2. It is helpful to highlight the mechanism of tirzepatide in treating metabolic diseases especially type 2 diabetes. It is also important to state the difference of tirzepatide with GLP-1 receptor agonists such as semaglutide as well as other dual or triple agonists. 

3. It is also important to compare tirzepatide with other diabetes treatment drugs. A recent large systematic review could be helpful (BMJ. 2023 Apr 6;381:e074068). 

4. It is critical to discuss the quality of life changes related to tirzepatide. 

Comments on the Quality of English Language

The English language is generally OK. 

Author Response

Dear Reviewer, 

Thanks for your comments and suggestions. Please find a point-by-point reply to your question.

  1. "Tirzeptide is a GLP-1/GIP dual agonist, but not a GLP-1 receptor agonist. Please avoid the confusion regarding the two drug classes."

It is correct. TZP is a dual (GIP-GLP-1) receptor agonist, as reported in the background (text in red).

    2. "It is helpful to highlight the mechanism of tirzepatide in treating metabolic diseases especially type 2 diabetes. It is also important to state the difference of tirzepatide with GLP-1 receptor agonists such as semaglutide as well as other dual or triple agonists."

It is correct. TZP is a dual (GIP-GLP-1) receptor agonist, as reported in the background and remarked in the conclusion (check the text for details; changes are in red).

    3. "It is also important to compare tirzepatide with other diabetes treatment drugs. A recent large systematic review could be helpful (BMJ. 2023 Apr 6;381:e074068)."

Thanks for your appropriate comment. The reference you indicated has been read, discussed, and cited in the paper (reference n 21). Please find the specific adjustment in the "Tirzepatide and body weight" section.

     4. "It is critical to discuss the quality of life changes related to tirzepatide."

Thanks for the suggestion. You can find the adjustment in the conclusion (text in red).

Round 2

Reviewer 1 Report

Comments and Suggestions for Authors

Table 1 is much more clear now. Furthermore Table 1 &2 are shown in a similar way whicch is more undertdable.

Comments on the Quality of English Language

None, english is fine to me.

Reviewer 2 Report

Comments and Suggestions for Authors

Thanks for the amends from the authors, which greatly improved the manuscript.